# DeepXML: Scalable & Accurate Deep Extreme Classification for Matching User Queries to Advertiser Bid Phrases

## Abstract

The objective in deep extreme multi-label learning is to jointly learn feature representations and classifiers to automatically tag data points with the most relevant subset of labels from an extremely large label set. Unfortunately, state-of-the-art deep extreme classifiers are either not scalable or inaccurate for short text documents. This paper develops the DeepXML algorithm which addresses both limitations by introducing a novel architecture that splits training of head and tail labels. DeepXML increases accuracy by (a) learning word embeddings on head labels and transferring them through a novel residual connection to data impoverished tail labels; (b) increasing the amount of negative training data available by extending state-of-the-art negative sub-sampling techniques; and (c) re-ranking the set of predicted labels to eliminate the hardest negatives for the original classifier. All of these contributions are implemented efficiently by extending the highly scalable Slice algorithm for pretrained embeddings to learn the proposed DeepXML architecture. As a result, DeepXML could efficiently scale to problems involving millions of labels that were beyond the pale of state-of-the-art deep extreme classifiers as it could be more than 10x faster at training than XML-CNN and AttentionXML. At the same time, DeepXML was also empirically determined to be up to 19% more accurate than leading techniques for matching search engine queries to advertiser bid phrases. Source code for DeepXML can be downloaded from (Anonymous, 2019).

## 1 Introduction

**Objective**: This paper develops the DeepXML algorithm for deep extreme multi-label learning applied to short text documents such as web search engine queries. DeepXML is demonstrated to be significantly more accurate and an order of magnitude faster to train than state-of-the-art deep extreme classifiers XML-CNN (Liu et al., 2017) and AttentionXML (You et al., 2018). As a result, DeepXML could efficiently train on problems involving millions of labels on a single GPU that were beyond the scaling capabilities of leading deep extreme classifiers. This allowed DeepXML to be applied to the problem of matching millions of advertiser bid phrases to a user's query on a popular web search engine where it was found to increase prediction accuracy by more than 19 percentage points as compared to the leading techniques currently in production.

**Deep extreme multi-label learning**: The objective in deep extreme multi-label learning is to learn feature representations and classifiers to automatically tag data points with the most relevant *subset* of labels from an extremely large label set. Note that multi-label learning is a generalization of multi-class classification which aims to predict a single mutually exclusive label.

**Notation**: Throughout the paper: $N$ refers to number of training points, $d$ refers to representation dimension, and $L$ refers to number of labels. Additionally, $\mathbf{Y}$ refers to the label matrix where $y_{ij} = 1$ if $j^{th}$ label is relevant to $i^{th}$ instance, and 0 otherwise. Please note that differences in accuracies are reported in absolute percentage points unless stated otherwise.

**Matching queries to bid phrases**: Web search engines allow ads to be served for not just queries bidded on directly by advertisers, referred to as bid phrases, but also for related queries with matching intent. Thus matching a query that was just entered by the user to the relevant subset of millions of

advertiser bid phrases in milliseconds is an important research application which forms the focus of this paper. DeepXML reformulates this problem as an extreme multi-label learning task by treating each of the top 3 Million monetizable advertiser bid phrases as a separate label and learning a deep classifier to predict the relevant subset of bid phrases given an input query. For example, given the user query "what is diabetes type 2" as input, DeepXML predicts that ads corresponding to the bid phrases "what is type 2 diabetes mellitus", "diabetes type 2 definition", "do i have type 2 diabetes", *etc.* could be relevant to the user. Note that other high-impact applications have also been reformulated as the extreme classification of short text documents such as queries, webpage titles, *etc.* For instance, (Jain et al., 2019) applied extreme multi-label learning to recommend the subset of relevant Bing queries that could be asked by a user instead of the original query. Similarly, extreme multi-label learning could be used to predict which subset of search engine queries might lead to a click on a webpage from its title alone for scenarios where the webpage content might not be available due to privacy concerns, latency issues in fetching the webpage, *etc*.

**State-of-the-art extreme classifiers**: Unfortunately, state-of-the-art extreme classifiers are either not scalable or inaccurate for queries and other short text documents. In particular, leading extreme classifiers based on bag-of-words (BoW) features (Prabhu et al., 2018b) and pretrained embeddings (Jain et al., 2019) are highly scalable but inaccurate for documents having only 3 or 4 words. While feature engineering (Arora, 2017; Joulin et al., 2017; Wieting & Kiela, 2019), including taking sub-word tokens, bigram tokens, *etc* can ameliorate the problem somewhat, their accuracy still lags that of deep learning methods which learn features specific to the task at hand. However, such methods, as exemplified by the state-of-the-art XML-CNN (Liu et al., 2017) and AttentionXML (You et al., 2018), can have prohibitive training costs and have not been shown to scale beyond a million labels on a single GPU. At the same time, there is a lot of scope for improving accuracy as XML-CNN and AttentionXML's architectures have not been specialized for short text documents.

**Tail labels**: It is worth noting that all the computational and statistical complexity in extreme classification arises due to the presence of millions of tail labels each having just a few, often a single, training point. Such labels can be very hard to learn due to data paucity. However, in most applications, predicting such rare tail labels accurately is much more rewarding than predicting common and obvious head labels. This motivates DeepXML to have specialized architectures for head and tail labels which lead to accuracy gains not only in standard metrics which assign equal weights to all labels but also in propensity scored metrics (Jain et al., 2016) designed specifically for long-tail extreme classification.

**DeepXML**: DeepXML improved both accuracy and scalability over existing deep extreme classifiers by partitioning all $L$ labels into a small set of head labels, with cardinality less than $0.1L$, containing the most frequently occuring labels and a large set of tail labels containing everything else. DeepXML first represented a document by the tf-idf weighted linear combination of its word-vector embeddings as this architecture was empirically found to be more suitable for short text documents than the CNN and attention based architectures of XML-CNN and AttentionXML respectively. The word-vector embeddings of the training documents were learnt on the head labels where there was enough data available to learn a good quality representation of the vocabulary. Accuracy was then further boosted by the introduction of a novel residual connection to fine-tune the document representation for head labels. This head architecture could be efficiently learnt on a single GPU with a fully connected final output layer due to the small number of labels involved. The word-vector embeddings were then transferred to the tail network where there wasn't enough data available to train them from scratch. Accuracy gains could potentially be obtained by fine tuning the embeddings but this led to a dramatic increase in the training and prediction costs. As an efficient alternative, DeepXML achieved state-of-the-art accuracies by fine tuning only the residual connection based document representation for tail labels. A number of modifications were made to the highly scalable Slice classifier (Jain et al., 2019) for pre-trained embeddings to allow it to also train the tail residual connection without sacrificing scalability. Finally, instead of learning an expensive ensemble of base classifiers to increase accuracy (Prabhu et al., 2018b; You et al., 2018), DeepXML improved performance by re-ranking the set of predicted labels to eliminate the hardest negatives for the base classifier with only a $10\%$ increase in training time.

**Results**: Experiments on medium scale datasets of short text documents with less than a million labels revealed that DeepXML's accuracy gains over XML-CNN and AttentionXML could be up to 3.92 and 4.32 percentage points respectively in terms of precision@$k$ and up to 5.32 and 4.2 percentage points respectively in terms of propensity-scored precision@$k$. At the same time, DeepXML could

be up to $15\times$ and $41\times$ faster to train than XML-CNN and AttentionXML respectively on these datasets using a single GPU. Furthermore, XML-CNN and AttentionXML were unable to scale to a proprietary dataset for matching queries to bid phrases containing 3 million labels and 21 million training points on which DeepXML trained in 14 hours on a single GPU. On this dataset, DeepXML was found to be at least 19 percentage points more accurate than Slice, Parabel (Prabhu et al., 2018b), and other leading query bid phrase-matching techniques currently running in production.

**Contributions**: This paper makes the following contributions: (a) It proposes the DeepXML architecture for short text documents that is more accurate than state-of-the-art extreme classifiers; (b) it proposes an efficient training algorithm that allows DeepXML to be an order of magnitude more scalable than leading deep extreme classifiers; and (c) it demonstrates that DeepXML could be significantly better at matching user queries to advertiser bid phrases as compared to leading techniques in production on a popular web search engine. Source code for DeepXML and the short text document datasets used in this paper can be downloaded from (Anonymous, 2019).

## 2   RELATED WORK

Much work has been done in extreme multi-label classification which can be broadly categorized in two categories: a) learning the classifier with pre-computed features (Agrawal et al., 2013; Cissé et al., 2013; Prabhu & Varma, 2014; Yu et al., 2014; Weston et al., 2013; Mineiro & Nikos, 2014; Jain et al., 2016; 2017; 2019; Yen et al., 2016; 2018; Xu et al., 2016; Babbar & Schölkopf, 2017; 2019; Niculescu-Mizil & Abbasnejad, 2017; Papanikolaou & Tsoumakas, 2017; Prabhu et al., 2018a;b; Barezi et al., 2019; Jasinska et al., 2016), b) jointly learning feature representation along with the classifier (Chen & Lin, 2012; Balasubramanian & Lebanon, 2012; Bi & Kwok, 2013; Liu et al., 2017; Jernite et al., 2017; Wydmuch et al., 2018; Bhatia et al., 2015; Tagami, 2017; You et al., 2018; Krichene et al., 2019; Barezi et al., 2019). More extensive survey of extreme classification and deep learning approaches can be found in section A.5 in the supplementary material.

Traditionally, extreme classification approaches used sparse BoW features due to fast & efficient feature computation as well as state-of-the-art performance for a large number of labels. However, for short text documents such as queries, which form the focus of the paper, deep learning representations are more effective than BoW (Jain et al., 2019). Unfortunately, existing deep learning approaches for extreme classification yielding state-of-the-art accuracy are not scalable, while scalable approaches have not been shown to yield state-of-the-art accuracy. In particular, XML-CNN, acheives state-of-the-art accuracy on short text documents but have not been shown to scale beyond a million labels, whereas AttentionXML was found to be slightly more scalable but less accurate as shown in table 1.

**Scalability:** Deep learning techniques have been very successful and comprehensively beaten BoW features in small output space (Kim, 2014; Yang et al., 2016). Unfortunately, the scalablity of such techniques degrades in extreme multi-label (XML) setting as the final fully connected layer leads to cost linear in number of labels (Liu et al., 2017). Traditional approach to solve this problem is negative sampling (Mikolov et al., 2013). Unfortunately, at the extreme scale, negative sampling has to be applied more aggressively as demonstrated in Fig 5 in the appendix. To eliminate this problem, many approaches have been proposed such as tree based (Prabhu et al., 2018b; You et al., 2018; Mikolov et al., 2013; Jernite et al., 2017) and hashing based (Shrivastava & Li, 2014; Vijayanarasimhan et al., 2014), approximate nearest neighbor sub-sampling techniques (Jain et al., 2019; Reddi et al., 2018). Adding to that, prediction time complexity remains $O(dL)$, which is not suitable for real-time predictions. Approaches such as Maximum Inner Product Search (Yen et al., 2018) and Local Sensitive Hashing (Niculescu-Mizil & Abbasnejad, 2017) can speed-up one-vs.-all inference but it leads to further loss in accuracy when applied post-training.

**Accuracy:** Tail labels are harder to predict as compared to head labels due to training data scarcity but might also be more informative and rewarding in XML setting. For instance, predicting tag "Artificial intelligence researchers" could be more informative than tag "1954 deaths" on Wikipedia page of "Alan Turing". Hence, propensity based precision and nDCG are the focus of the paper. (Wei & Li, 2018) demonstrates that trimming tail labels leads to marginal decay in performance on vanilla metrics. However, a) trimming tail labels lead to loss in propensity based metrics, b) this approach has been demonstrated to scale to $\approx 30K$ labels only. Researchers have also tried to improve performance on tail labels by directly optimizing propensity scored metrics (Jain et al., 2016), posing learning in the presence of adversarial perturbations (Babbar & Schölkopf, 2019) and treating tail labels as outliers (Xu et al., 2016). Although, these approaches boost performance on tail labels, however they are not well suited for short text documents due to, a) support only for fixed

features and, b) large training and prediction time.

**Matching user queries to bid phrases:** Approaches in this domain can be categorized as embeddings based (Jain et al., 2019), sequence-to-sequence models (Gao et al., 2012; Jones et al., 2006; Riezler & Liu, 2010; He et al., 2016; Lee et al., 2018; Lian et al., 2019) and query graph based models (Ioannis et al., 2008). Unfortunately, the trigger coverage, suggestion density and quality of recommendations could be poor for many of these techniques. For instance, query graph based methods can only recommend suggestions for previously seen triggers thereby limiting their trigger coverage. Additionally, Sequence-to-sequence models suffers from expensive training and prediction cost. Although, efficient structures such as trie (Lian et al., 2019) have been deployed to reduce output complexity, however, at the cost of limited bid phrase coverage.

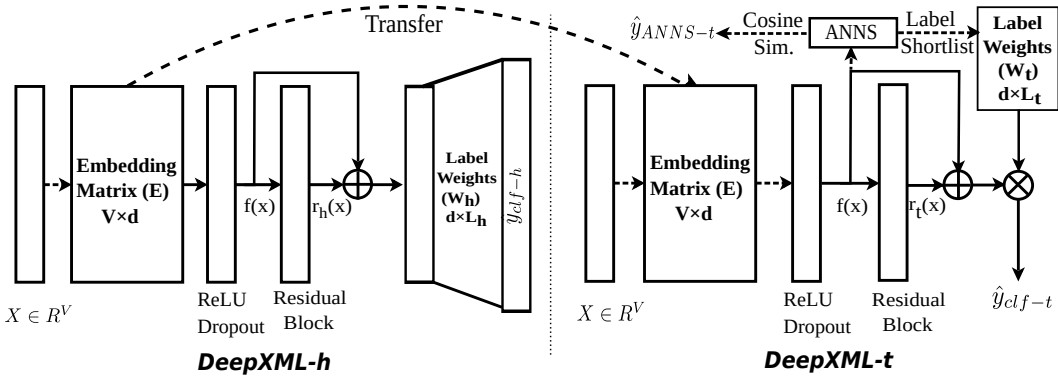

**Figure 1:** Architecture of DeepXML with DeepXML-h for head labels and DeepXML-t for tail labels. Residual block is a sequence of linear transformation, Batch Normalization, ReLU and Dropout operations. Here, $\otimes$ refers to matrix-vector multiplication. Gradients are not propagated through dotted lines. Please refer to Fig. 6 for DeepXML's architecture with an intuitive example.

## 3 DEEPXML

State-of-the-art deep extreme classifiers are neither scalable nor accurate for short text documents because: (a) representations learnt through CNNs (XML-CNN), LSTM+Attention (AttentionXML) or context (Bert (Devlin et al., 2018), Elmo (Peters et al., 2018), *etc.*) might not be accurate given limited tail data with only a few words per document and might also be expensive to compute; and (b) they face scalability issues as the final fully connected output layer has millions of outputs making both the forward pass as well as gradient backpropagation prohibitive for even a single training point. DeepXML addresses these limitations by using a feature representation inspired by FastText (Joulin et al., 2017) and an output layer inspired by Slice (Jain et al., 2019) as these have been demonstrated to be both accurate and scalable for short text documents in their individual capacities. Unfortunately, combining FastText and Slice in a straight forward fashion also turns out to be neither accurate nor scalable. The rest of this section details the design choices and modifications that needed to be made in order to get the combination to work.

**DeepXML, FastText & Slice**: FastText represents a document as an efficiently computable linear combination of its (sub) word-vector embeddings making it highly scalable and well suited for extreme classification scenarios. Unfortunately, the FastText architecture is linear and low-capacity thereby leading to a loss in accuracy when there isn't a vast amount of unsupervised data available for training. Furthermore, previous attempts (Wydmuch et al., 2018) at learning FastText representations in a supervised manner by replacing the fully connected output layer by a fixed Probabilistic Label Trees (PLT) extreme classifier (Jasinska et al., 2016) have led to even worse accuracies than XML-CNN and AttentionXML. Replacing the fixed tree PLT by learnt Slice improves accuracy somewhat but does not lead to state-of-the-art results and also greatly increases training time (please see ablation experiments in Section 5). DeepXML addresses these limitations by adding a non linearity and residual block to make up for the lack of FastText's expressive power and training the word-vector embeddings using a fully connected output layer on a small number of head labels rather than through Slice. Please refer to Figure 1 for full architecture of DeepXML. Once the word-vector embeddings have been trained on the head, they are frozen and transferred to the tail where only the residual

block is fine tuned. This increases both accuracy as there isn't enough training data available to learn good quality word embeddings from scratch on the tail as well as scalability as fine-tuning the head embeddings on the tail would prove too expensive on large problems.

### 3.1 ARCHITECTURE AND TRAINING

**DeepXML-h**: Model parameters *i.e.* words embeddings, residual block and fully connected classifier are learnt with Binary Cross Entropy loss and Adam optimizer. DeepXML-h can be efficiently trained with a fully connected final layer on a single GPU, as $L_h$ (Label set containing head labels)contains only a small subset of labels. In practice, the size of $L_h$ does not grow beyond 0.2M even for datasets with millions of labels. Additionally, an approximate nearest neighbour search (ANNS) structure was trained over label centroids $\mathbf{L}_h^c$ of head labels. Note that label centroids ($\mathbf{L}_h^c$) are computed as: $\mathbf{L}_h^c = \mathbf{Y}^T \mathbf{X}'$, please note that $\mathbf{X}' = \mathbf{f}(\mathbf{x}) + \mathbf{r_h}(\mathbf{x})$ from the Fig. 1. This brings down inference cost of classifier from $O(d|L_h|)$ to $O(d|s| + d \log|L_h|)$, given $|s| \ll |L_h|$. Here, $|s|$ is the size of label shortlist queried from ANNS structure during prediction which is kept as 300 in practice.
Unfortunately, ANNS trained over label centroids may lead to poor recall values when a single centroid is unable to capture diversity in training instances for labels with highly different contexts (say $L_h'$). For instance, 280K articles, ranging from scientists to athletes, are tagged with the 'living people' tag in WikiTitle-500K (Bhatia et al., 2016) dataset. Slice increases the shortlist to improve recall; however, at the cost of $6\times$ increase in prediction time. DeepXML-h tackles this issue by allowing multiple representations for labels in $L_h'$. Documents are clustered using the KMeans algorithm into $c$ clusters for each label in this set. Therefore, $c$ representatives for each label in $L_h'$ are computed. This leads to 5% increase in recall@300 and 6% precision@300 with a shortlist of size 300. Clustering will lead to $|L_h - L_h'| + c|L_h'|$ label representations and hence could potentially lead to increased time complexity. However, clustering just the top 3 labels into 300 clusters seems to work well in our experiments without significant change in training time, and it doesn't impact prediction time at all.
**DeepXML-t**: DeepXML still relies on Slice to fine-tune the residual block in the tail network and learn the weights in the fully connected output layer for millions of tail labels. Slice cuts down the time for both the forward pass and the gradient backpropagation from linear to logarithmic in the number of labels. Slice achieves this by first representing each label by the unit normalized mean of the feature vectors of the training points tagged with the label. It then learns an ANNS data structure (Malkov & Yashunin, 2016) over the label representations to determine the most likely labels for a given data point (please see the supplementary material for a more detailed description). This technique was shown to efficiently scale to millions of labels and training points when the feature representation was fixed (Jain et al., 2019). Unfortunately, when the feature representation is being learnt, the ANNS data structure needs to be constantly updated as the label representations change with each training batch. This can lead to a marked slowdown as maintaining the ANNS data structure can incur significant computational cost.

DeepXML speeds up training by redefining the label representation to be the unit normalized mean of the document representation before the residual block computed using the learnt word embeddings alone. This allows the ANNS data structure to be learnt just once after the word embeddings have been learnt on the head labels and before training starts on the tail. Unfortunately, this also leads to a loss in training accuracy as the label representation is now an approximation to the true representation that should have been defined as the unit normalized mean of the document representations computed after the residual block. This loss in accuracy can be compensated by requiring Slice to generate 3x more nearest labels to a given data point. This allowed the hard negative labels to now be present in the longer list but this significantly increased training time. It was empirically determined that a more efficient strategy was to extend the shortlist by adding randomly sampled negative labels as this led to no loss in accuracy with only a minimal increase in training time.

### 3.2 PREDICTION

Initially, classifier scores and ANNS scores from both DeepXML-h and DeepXML-t are merged in a single vector $\hat{\mathbf{y}}$. As demonstrated in Fig. 1, $\hat{\mathbf{y}}_{clf-h}$ is the classifier score (logit) from DeepXML-h and $\hat{\mathbf{y}}_{anns-h}$ is the cosine similarity from ANNS-h. Similarly, $\hat{\mathbf{y}}_{clf-t}$ is the classifier score from DeepXML-t and $\hat{\mathbf{y}}_{ann-t}$ is the cosine similarity from ANNS-t. The final DeepXML score vector ($\hat{\mathbf{y}}$)

is computed as follows :

$$\hat{\mathbf{y}} = (1 - \beta) * \sigma([\hat{\mathbf{y}}_{anns-h}; \hat{\mathbf{y}}_{anns-t}]) + \beta * \sigma([\hat{\mathbf{y}}_{clf-h}; \hat{\mathbf{y}}_{clf-t}]) \tag{1}$$

Note that $\hat{\mathbf{y}}_{clf-t}$, $\hat{\mathbf{y}}_{clf-h}$, $\hat{\mathbf{y}}_{anns-h}$, and $\hat{\mathbf{y}}_{anns-t}$ are sparse vectors, $\sigma$ is a sparse-sigmoid function computed only at non-zero entries and $\beta \in [0, 1]$. The average cost of prediction can be broken down into the following four components: a) computing dense feature representation: $O(d\gamma)$, b) generating a shortlist: $O(d \log |L_t|)$, c) computing classifier scores: $O(d|s|)$. Here, $\gamma$ is the average number of features per document and $|s|$ is the shortlist size in ANNS.

### 3.3 DEEPXML-RE

Extreme classifiers such as Parabel (Prabhu et al., 2018b), AttentionXML (You et al., 2018) learn multiple models to get better prediction accuracy. However, this leads to increased training and prediction time (200% for both Parabel and AttentionXML) linearly with the number of models. Whereas, DeepXML-RE learns a re-ranker with training cost logarithmic in the number of labels by training over a shortlist of negative labels. Specifically, false positive labels predicted by DeepXML (*a.k.a* hardest negatives) are selected as negatives for DeepXML-RE. This leads to only $10 - 20\%$ increase in training time. The architecture of DeepXML-RE is same as DeepXML-t, i.e., a word embedding layer, residual block, and classifier. Model parameters are learnt with binary cross entropy loss and SparseAdam optimizer. During prediction, DeepXML-RE evaluates on labels shortlisted by DeepXML only, thereby incurring a prediction cost of $O(d|s|)$.

## 4 EXPERIMENTS AND RESULTS

### 4.1 DATA SETS AND EVALUATION METRICS

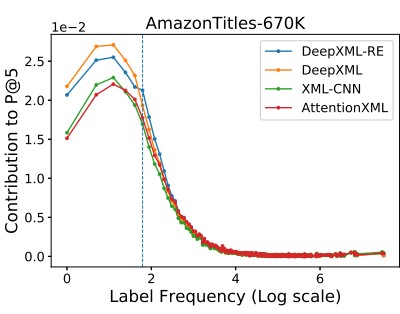

**Figure 2:** Contribution to P@5 by labels of a given frequency for AmazonTitles-670K. The dotted vertical blue line marks the threshold to the right of which are head labels, and left of which are tail labels.

Experiments were carried out on the Query to Bid phrases (Q2B-3M) dataset, with 3 million labels, by mining the logs of a popular search engine. Each user query was treated as an instance and relevant advertiser bid phrases became its labels. Unfortunately, only Slice (Jain et al., 2019) and Parabel (Prabhu et al., 2018b) could scale to this dataset. DeepXML is also compared to state-of-the-art methods for query keyword prediction such as Simrank++(Ioannis et al., 2008) and a sequence-to-sequence model based on BERT (Devlin et al., 2018; Lian et al., 2019).

Experiments were also carried out on four moderate size datasets (Anonymous (2019)). The applications considered were tagging Wikipedia pages (WikiTitles-500K), suggesting relevant articles (WikiSeeAlsoTitles-250K), and item-to-item recommendation of Amazon products (AmazonTitles-670K and AmazonTitles-3M). Please refer to supplementary Table 4 for dataset statistics. For these moderate size datasets DeepXML was compared to leading deep learning and BoW feature-based methods including XML-CNN, Attention-XML, Slice, AnnexML (Tagami, 2017), PfastreXML (Jain et al., 2016), Parabel (Prabhu et al., 2018b), XT (Wydmuch et al., 2018), and DiSMEC (Babbar & Schölkopf, 2017). The implementation of all algorithms was provided by the respective authors.

### 4.2 HYPERPARAMETERS

DeepXML has 7 hyperparameters: (a) learning rate and epochs for DeepXML-h; (b) learning rate and epochs for DeepXML-t; (c) embedding dimensions ($d$); (d) shortlist size ($|s|$) and (e) threshold to split label set into head and tail labels. Results are reported for $d = 300$ and $|s| = 300$. The label threshold is chosen via cross-validation (please refer to section A.2 in the supplementary material). Binary cross-entropy loss and the SparseAdam optimizer were used to update the model parameters. Please refer to Table 5 in the supplementary section for full parameter settings.

**Table 1:** DeepXML and DeepXML-RE are more accurate on both vanilla and propensity-scored metrics, and are faster at training. '-' is used for the methods which are unable to scale. * is used for the results provided by respective authors.

| Dataset | Method | P@1 | P@3 | P@5 | N@5 | PSP@1 | PSP@3 | PSP@5 | PSN@5 | Training Time (hrs) | Prediction Time (ms) | Model Size (GB) |
|---|---|---|---|---|---|---|---|---|---|---|---|---|
| WikiSeeAlsoTitles-350K | DeepXML | 19.62 | 13.76 | 10.79 | 19.73 | **10.43** | **12.4** | **14.1** | **13.23** | 0.94 | 1.03 | 1.02 |
| | DeepXML-RE | **20.18** | **14.04** | 10.99 | **20.05** | 9.95 | 11.87 | 13.51 | 12.64 | 1.09 | 1.52 | 1.52 |
| | DeepXML-fr | 19.82 | 13.95 | **11.02** | 19.87 | 9.26 | 11.25 | 13.01 | 11.98 | 3.91 | 0.48 | 0.44 |
| | XML-CNN | 17.75 | 12.34 | 9.73 | 17.48 | 8.24 | 9.72 | 11.15 | 10.31 | 14.25 | 5.09 | 0.78 |
| | XT | 16.55 | 11.37 | 8.93 | 16.47 | 7.38 | 8.75 | 10.05 | 9.46 | 3.25 | 4.53 | 2 |
| | SLICE+FastText | 18.13 | 12.87 | 10.29 | 18.52 | 8.63 | 10.78 | 12.74 | 11.63 | 0.22 | 1.33 | 0.97 |
| | AttentionXML-I | 14.85 | 9.51 | 7.2 | 13.6 | 5.76 | 6.31 | 7.04 | 6.56 | 5.1 | 2.07 | 0.7 |
| | AttentionXML | 15.86 | 10.43 | 8.01 | 14.86 | 6.39 | 7.2 | 8.15 | 7.64 | 30.44 | 14.88 | 4.07 |
| | DiSMEC | 16.61 | 11.57 | 9.14 | 16.72 | 7.48 | 9.19 | 10.74 | 9.99 | 6.63 | 7.63 | 0.09 |
| | Parabel | 17.24 | 11.61 | 8.92 | 16.67 | 7.56 | 8.83 | 9.96 | 9.45 | 0.06 | 1.14 | 0.43 |
| | AnnexML | 14.96 | 10.2 | 8.11 | 14.76 | 5.63 | 7.04 | 8.59 | 7.76 | 0.2 | 0.1 | 3.59 |
| | PfastreXML | 15.09 | 10.49 | 8.24 | 15.59 | 9.03 | 9.69 | 10.64 | 10.52 | 0.51 | 3.17 | 5.22 |
| Wikipedia-Title-500K | DeepXML | 44.57 | 24.04 | 16.86 | 30.93 | **18.82** | 17.92 | 17.96 | 20.88 | 3.7 | 2.64 | 1.7 |
| | DeepXML-RE | 45.26 | 24.34 | 16.94 | 31.21 | 18.76 | **18.1** | 18.06 | **21** | 4.63 | 3.49 | 2.47 |
| | DeepXML-fr | **45.67** | **25.59** | **18.23** | **32.67** | 17.86 | 17.81 | **18.07** | 20.7 | 16.72 | 0.74 | 0.76 |
| | XML-CNN | 43.45 | 23.24 | 16.53 | 29.95 | 15.64 | 14.74 | 14.98 | 17.45 | 55.22 | 6.49 | 1.17 |
| | XT | 39.44 | 21.57 | 15.31 | 27.65 | 15.23 | 15 | 15.25 | 17.59 | 3.89 | 3.89 | 3.3 |
| | SLICE+FastText | 28.07 | 16.78 | 12.28 | 22.87 | 15.1 | 14.69 | 15.33 | 17.67 | 0.55 | 1.11 | 1.5 |
| | AttentionXML-1 | 41.75 | 21.49 | 14.86 | 27.38 | 14.56 | 13.25 | 12.98 | 15.53 | 34.15 | 2.93 | 3.07 |
| | AttentionXML | 42.89 | 22.71 | 15.89 | 28.93 | 15.12 | 14.32 | 14.22 | 16.75 | 102.43 | 8.79 | 9.21 |
| | DiSMEC | 39.89 | 21.23 | 14.96 | 27.32 | 15.89 | 15.15 | 15.43 | 17.86 | 23.95 | 8.87 | 0.35 |
| | Parabel | 42.5 | 23.04 | 16.21 | 29.45 | 16.55 | 16.12 | 16.16 | 18.77 | 0.34 | 0.89 | 2.15 |
| | AnnexML | 39.56 | 20.5 | 14.32 | 26.54 | 15.44 | 13.83 | 13.79 | 18.46 | 1.78 | 0.09 | 10.7 |
| | PfastreXML | 30.99 | 18.07 | 13.09 | 23.88 | 17.87 | 15.4 | 15.15 | 18.46 | 3.07 | 7.75 | 16.85 |
| Amazon-Titles-670K | DeepXML | 38.38 | 34.32 | 31.25 | 35.11 | **27.63** | **29.27** | 30.83 | **28.98** | 1.42 | 1.12 | 2.09 |
| | DeepXML-RE | 39 | 34.93 | 31.85 | 35.72 | 27.31 | 29.22 | **30.94** | 28.96 | 1.55 | 1.39 | 2.92 |
| | DeepXML-fr | 38.42 | 34.47 | 31.44 | 35.21 | 25.54 | 27.91 | 29.94 | 27.76 | 6.3 | 1 | 0.82 |
| | XML-CNN | 35.02 | 31.37 | 28.45 | 31.94 | 21.99 | 24.93 | 26.84 | 24.67 | 23.52 | 8.8 | 1.36 |
| | XT | 36.57 | 32.73 | 29.79 | 33.35 | 22.11 | 24.81 | 27.18 | 24.87 | 4.66 | 8.52 | 4 |
| | SLICE+FastText | 33.85 | 30.07 | 26.97 | 30.56 | 21.91 | 24.15 | 25.81 | 24.23 | 0.22 | 1.59 | 2.01 |
| | AttentionXML-1 | 36.08 | 31.68 | 28.3 | 32.04 | 23.03 | 24.63 | 25.98 | 24.33 | 12.79 | 5.87 | 4.04 |
| | AttentionXML | 37.92 | 33.73 | 30.57 | 34.35 | 24.24 | 26.43 | 28.39 | 26.33 | 37.5 | 15.9 | 12.11 |
| | DiSMEC* | **39.84** | **35.63** | **32.68** | **36.58** | 24.84 | 27.64 | 30.46 | - | - | - | - |
| | Parabel | 38 | 33.54 | 30.1 | 33.98 | 23.1 | 25.57 | 27.61 | 25.48 | 0.09 | 1.11 | 1.06 |
| | AnnexML | 35.31 | 30.9 | 27.83 | 31.26 | 17.94 | 20.69 | 23.3 | 20.88 | 0.17 | 0.1 | 2.99 |
| | PfastreXML | 32.88 | 30.54 | 28.8 | 31.85 | 26.61 | 27.79 | 29.22 | 27.59 | 0.99 | 9.31 | 5.32 |
| AmazonTitles-3M | DeepXML | 38.36 | 36.45 | 34.82 | 36.48 | 15.7 | 17.93 | 19.52 | 18.14 | 7.4 | 2.9 | 8.64 |
| | DeepXML-RE | 40.22 | 38.4 | 36.74 | 38.45 | 15.51 | 17.88 | 19.6 | 18.19 | 9.69 | 4.03 | 11.98 |
| | DeepXML-fr | - | - | - | - | - | - | - | - | - | - | - |
| | XML-CNN | - | - | - | - | - | - | - | - | - | - | - |
| | XT | - | - | - | - | - | - | - | - | - | - | - |
| | SLICE+FastText | 29.88 | 28.14 | 26.82 | 28.05 | 9.85 | 11.68 | 13.05 | 11.89 | 3.65 | 1.13 | 8.45 |
| | AttentionXML-1 | - | - | - | - | - | - | - | - | - | - | - |
| | AttentionXML | - | - | - | - | - | - | - | - | - | - | - |
| | DiSMEC* | 40.81 | **38.58** | **36.76** | **38.56** | - | - | - | - | - | - | - |
| | Parabel | **40.97** | 38.56 | 36.68 | 38.44 | 11.4 | 13.69 | 15.4 | 14 | 1.94 | 1.26 | 13.4 |
| | AnnexML | - | - | - | - | - | - | - | - | - | - | - |
| | PfastreXML | 27.58 | 27.76 | 27.5 | 28.37 | **20.02** | **21.95** | **23.29** | **21.99** | 12.56 | 22.55 | 23.3 |

### 4.3 RESULTS ON MODERATE SIZE DATA SETS

Table 1 compares the performance of DeepXML and DeepXML-RE with state-of-the-art deep learning approaches namely XML-CNN and AttentionXML. DeepXML is upto 5 percentage points more accurate than XML-CNN and AttentionXML on propensity-scored metrics. Additionally, Fig 2 and supplementary Fig 4 demonstrate that tail labels contribute more to DeepXML's predictions w.r.t other methods. Furthermore, DeepXML can be 2–4 percentage points more accurate than XML-CNN and AttentionXML on vanilla precision and nDCG. Table 1 also demonstrates that DeepXML can be 33–42× faster to train as compared to XML-CNN and AttentionXML respectively on a single GPU. DeepXML-RE can be scaled to AmazonTitles-3M on one GPU and is able to train with only a 10% increase in training time and a 20% increase in prediction time from DeepXML as compared to a 200% increase in training and prediction time of AttentionXML from AttentionXML-1. This demonstrates that DeepXML and DeepXML-RE can outperform current state-of-the-art accuracy as well as scale to millions of labels . Table 1 shows that DeepXML and DeepXML-RE can outperform state-of-the-art BoW-based approaches as well. Note that on AmazonTitles-3M, PfastreXML is 5.6 percentage points better than DeepXML as well as DeepXML-RE for propensity-scored metrics but incurs a loss of 10 percentage points for vanilla precision, which is clearly unacceptable for real world tasks where both metrics are important. Please refer to Section A.3 in the appendix for more discussion.

**Table 2:** Results on Q2B-3M. DeepXML-RE is 19% more accurate on P@1 than the second-best method.

| Method | P@1 | P@3 | P@5 | N@3 | N@5 | PSP@1 | PSP@3 | PSP@5 | PSN@3 | PSN@5 |
|---|---|---|---|---|---|---|---|---|---|---|
| DeepXML | 70.99 | 33.7 | **21.72** | 82.26 | 84.15 | 61.18 | 83.55 | 89.99 | 76.72 | 79.33 |
| DeepXML-RE | **73.37** | **33.91** | 21.67 | **83.52** | **85.13** | **64.8** | **85.13** | **90.42** | **79.1** | **81.24** |
| Parabel | 54.29 | 27.15 | 17.94 | 64.56 | 66.98 | 43.09 | 61.66 | 68.61 | 55.1 | 58.07 |
| Slice+CDSSM | 53.23 | 27.53 | 18.56 | 65.08 | 68.26 | 42.35 | 64.51 | 74.14 | 56.94 | 61.09 |
| Parabel-weighted | 43.82 | 22.25 | 14.97 | 52.15 | 54.61 | 37.83 | 53.19 | 59.57 | 47.26 | 49.94 |
| Seq2Seq | 28.25 | 13.06 | 8.02 | 22.96 | 21.7 | 23.32 | 17.59 | 15.42 | 21.31 | 21.3 |
| Simrank++ | 52.7 | 29.69 | 19.33 | 50.02 | 48.82 | 43.5 | 41.67 | 39.36 | 45.71 | 45.73 |

## 4.4 Results on Q2B

Table 2 indicates that DeepXML-RE is upto 20–45 percentage points more accurate than the methods currently in production, Simrank++ and Seq2Seq respectively. Further, DeepXML-RE is also compared to other scalable extreme classifiers such as Parabel and Slice and is seen to be 19 and 20 percentage points more accurate respectively. Additionally, supplementary Table 6 includes example documents which demonstrate that DeepXML can make accurate and diverse predictions.

## 5 Ablation, extensions

This section discuss impact of feature representations (DeepXML-SW, DeepXML-f) and classifiers (DeepXML-fr, DeepXML-ANNS, DeepXML-P, DeepXML-NS). The detailed architectures are included in section A.6 in the supplementary section.

**Sub-word features**: DeepXML can also exploit sub-word features (Joulin et al., 2017) for an additional 1% gain in precision as demonstrated in Table 3.

**Label split and classifier**: DeepXML-fr refers to a joint DeepXML architecture, *i.e.* without splitting labels, trained with a fully connected layer. As demonstrated in Table 1, the proposed algorithm is more scalable and accurate on tail labels w.r.t DeepXML-fr. DeepXML could be up to 1–3% more accurate then DeepXML-NS &

**Table 3:** Ablation results for different feature representations and classifiers on AmazonTitles-670K. DeepXML is more accurate than the alternate configurations.

| Method | P@1 | P@5 | PSP@1 | PSP@5 |
|---|---|---|---|---|
| DeepXML | 38.38 | 31.25 | 27.63 | 30.83 |
| DeepXML-RE | 39 | 31.85 | 27.31 | 30.94 |
| DeepXML-P | 37.07 | 28.75 | 23.29 | 26.54 |
| DeepXML-NS | 35.71 | 29.93 | 24.26 | 28.82 |
| DeepXML-f | 34.8 | 27.53 | 21.29 | 23.23 |
| DeepXML-SW | 39.04 | 31.93 | 28.6 | 31.72 |
| DeepXML-ANNS | 36.73 | 30.07 | 26.72 | 29.83 |
| Slice+BERT | 26.99 | 22.05 | 16.75 | 20.27 |
| Slice+SIF | 30.82 | 24.11 | 20.54 | 23.52 |

DeepXML-ANNS where the classifier is trained using negative sampling. Additionally, DeepXML was found to be 1.3% more accurate than DeepXML-P which uses a tree based classifier.

**Pre-trained features**: DeepXML could be upto 10% more accurate than pre-trained representations with Slice classifier (refer to Tables 1 & 3) such as FastText, BERT (Devlin et al., 2018) and SIF (Arora, 2017).

## 6 Conclusion

This paper developed DeepXML, an algorithm to jointly learn representations for extreme multi-label learning on text data. The proposed algorithm addresses the key issues of scalability and low accuracy (especially on tail labels and very short documents) with existing approaches such as Slice, AttentionXML, and XML-CNN, and hence improves on them substantively. Experiments revealed that DeepXML-RE can lead to a 1.0–4.3 percentage point gain in performance while being 33–42× faster at training than AttentionXML. Furthermore, DeepXML was upto 15 percentage points more accurate than leading techniques for matching search engine queries to advertiser bid phrases. We note that DeepXML's gains are predominantly seen to be on predicting tail labels (for which very few direct word associations are available at train time) and on short documents (for which very few words are available at test time). This indicates that the method is doing especially well, compared to earlier approaches, at learning word representations which allow for richer and denser associations between words – which allow for the words to be well-clustered in a meaningful semantic space, and hence useful and generalisable information about document labels extracted even when the number of direct word co-occurrences observed is very limited. In the future we would like to better understand the nature of these representations, and explore their utility for other linguistic tasks.

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

**Table 4:** Dataset Statistics. Please note that in Q2B-3M dataset, character 3-grams and 4-grams tokens were also included in the vocabulary for DeepXML, DeepXML-RE and Parabel. This increases average number of features per document.

| Dataset | Train Instances $N$ | Features $F$ | Labels $L$ | Number of Test Instances | Average Labels per sample | Average Points per labels | Average Features per instances |
|---|---|---|---|---|---|---|---|
| WikiSeeAlsoTitles-350K | 629418 | 91414 | 352072 | 162491 | 2.33 | 5.24 | 2.73 |
| WikiTitles-500K | 1699722 | 185479 | 501070 | 722678 | 4.89 | 23.62 | 2.73 |
| AmazonTitles-670K | 485176 | 66666 | 670091 | 150875 | 5.39 | 5.11 | 5.26 |
| AmazonTitles-3M | 1712536 | 165431 | 2812281 | 739665 | 36.18 | 31.55 | 6.83 |
| Q2B-3M | 21561529 | 1284191 | 3192113 | 6995038 | 1.2 | 10.72 | 33.68 |

# A  SUPPLEMENTARY

## A.1  HYPERPARAMETERS OF DEEPXML FOR REPRODUCIBILITY

Table 5 lists the parameter settings for different data sets. Experiments were performed with a random-seed of 22 on a P40 GPU card with CUDA 10, CuDNN 7.4, and Pytorch 1.2 (Paszke et al., 2017).

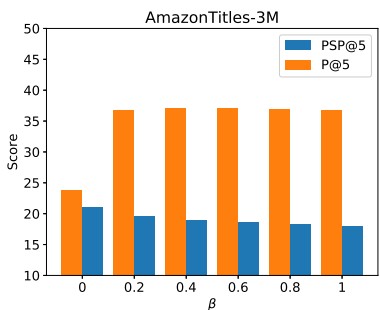

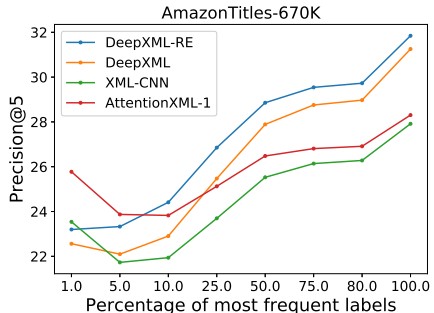

**Figure 3:** Variation of scores (Precision and Propensity scored Precision) with $\beta$

**Figure 4:** Precision@5 in k(%) most frequent labels

**Table 5:** Parameter setting for DeepXML on different datasets. Dropout with probability 0.5 was used for all datasets. Learning rate is decayed by Decay factor after interval of Decay steps. For HNSW, values of construction parameter $M = 100$, $efC = 300$ and query parameter, $efS = 300$. Denoted by '|', DeepXML-h and DeepXML-t might take different values for some parameters. Note that DeepXML-t uses a shortlist of size 500 during training. However, a shortlist of size 300 queried from ANNS is used at prediction time for both DeepXML-h and DeepXML-t.

| Dataset | $|L_h|$ | $|L_t|$ | Epochs | Learning Rate | Batch Size | Decay Factor | Decay Steps | Label Threshold | Shortlist Size |
|---|---|---|---|---|---|---|---|---|---|
| WikiSeeAlsoTitles-350K | 52,245 | 252,205 | 25 \| 30 | 0.005 \| 0.002 | 255 | 0.5 | 14,7 \| 10 | 5 | 300\|500 |
| WikiTitles-500K | 58,476 | 43,7845 | 25 \| 20 | 0.005 \| 0.002 | 255 | 0.5 | 14,7 \| 10 | 25 | 300\|500 |
| AmazonTitles-670K | 109,379 | 556,696 | 25 \| 20 | 0.02 \| 0.002 | 255 | 0.5 | 14,7 | 6 | 300\|500 |
| AmazonTitles-3M | 165,722 | 2,645,458 | 25 \| 20 | 0.007 \| 0.002 | 255 | 0.5 | 14,7 \| 10 | 75 | 300\|500 |
| Q2B-3M | 160,651 | 3,019,249 | 8 \| 5 | 0.02 \| 0.002 | 512 | 0.5 | 4 \| 2 | 25 | 300\|500 |

## A.2  SPLITTING $L$ INTO $L_h$ AND $L_t$

Label set $L$ is divided into two disjoint sets, i.e. $L_h$ and $L_t$ based on the frequency of the labels. Labels with a frequency more than splitting threshold $\gamma$ are kept in set $L_h$ and others in $L_t$. The splitting threshold $\gamma$ is chosen while ensuring that most of the features (or words) are covered in documents that one at least one instances of label in the set $L_h$ and $|L_h| < 0.2M$. Two components for DeepXML, DeepXML-h and DeepXML-t, are trained on $L_h$ and $L_t$. Please note that other strategies like clustering of labels, connected components of labels in a graph were also tried, but the above-mentioned strategy provides good results without any additional overhead. More sophisticated algorithms for splitting such as label clustering, may yield better results, however at the cost of increased training time.

**Table 6:** Qualitative comparision for DeepXML on WikiTitles-500K dataset. Bold indicates correct predictions.

| Document Text | Ground Truth | DeepXML-re | DeepXML | XML-CNN | AttentionXML |
|---|---|---|---|---|---|
| Confederate Secret Service | 1861 establishments in the Confederate States of America, 1865 disestablishments in the Confederate States of America, American Civil War espionage, Defunct intelligence agencies, Government of the Confederate States of America, Military history of the Confederate States of America, Military units and formations established in 1861 | **1865 disestablishments in the Confederate States of America, Government of the Confederate States of America, 1861 establishments in the Confederate States of America**, Economic history of the Confederate States of America, **Military history of the Confederate States of America** | **1865 disestablishments in the Confederate States of America, Government of the Confederate States of America, 1861 establishments in the Confederate States of America**, Economic history of the Confederate States of America, Confederate States of America monuments and memorials | **1865 disestablishments in the Confederate States of America, 1861 establishments in the Confederate States of America**, American films, English-language films, Black-and-white films | American films, English-language films, Military history of the **Confederate States of America**, 2011 television episodes, English-language television programming |

**Table 7:** Qualitative comparison for DeepXML on Q2B-3M dataset. Bold indicates correct predictions. Please note that system in production filters label internally and then outputs them.

| Document text | Ground Truth Labels |
|---|---|
| what is diabetes type 2;what is type 2 diabetic | diabetes type 1 and 2;type diabetes 1 and 2;type 1 and 2 diabetes, what causes type 2 diabetes, diabetes type 2 definition;2 definition diabetes type;definition diabetes type 2;definition type 2 diabetes;diabetes definition type 2;type 2 diabetes definition, do i have type 2 diabetes, what is a type 2 diabetes, what are type 2 diabetes, what is type 2 diabetes mellitus, type 2 diabetes |
| **Method** | **Predicted labels** |
| Seq2Seq | what is type ii diabetes, "whats type 2 diabetes" |
| SimRank++ | **type 2 diabetes** |
| Parabel | what is type 2 diabetes, **type 2 diabetes**, what is type 2 diabetes symptoms, diabetes 2;2 diabetes;diabetes 2, whats type 2 diabetes, what is email marketing, what is type 1 and type 2 diabetes, what are the symptoms of diabetes type 2, what is anemia;anemia is what;anemia what is;what anemia is,what is psd2, what identity theft is, what is google adwords, what is radiation therapy, what is emilys list, what is pork belly, what is qualcomm, what is a smartboard, what is an unlocked phone, mpls, what is the invoice price |
| DeepXML | what is type 2 diabetes, **what is a type 2 diabetes**, **type 2 diabetes**, **what are type 2 diabetes**, what is the type 2 diabetes, **do i have type 2 diabetes**, "whats type 2 diabetes", **what is type 2 diabetes mellitus, diabetes type 2 definition;2 definition diabetes type;definition diabetes type 2;definition type 2 diabetes;diabetes definition type 2;type 2 diabetes definition**, what is type 1 and type 2 diabetes |
| DeepXML-RE | what is type 2 diabetes, **type 2 diabetes**, **what is a type 2 diabetes**, **do i have type 2 diabetes**, what is the type 2 diabetes, **what are type 2 diabetes**, "what's type 2 diabetes", **what is type 2 diabetes mellitus, diabetes type 2 definition;2 definition diabetestype;definition diabetes type 2;definition type 2 diabetes;diabetes definition type 2;type 2 diabetes definition**, what is type 1 and type 2 diabetes, what are the causes of diabetes type 2, types of diabetes 2, what is diabetes 2, **what causes type 2 diabetes** |

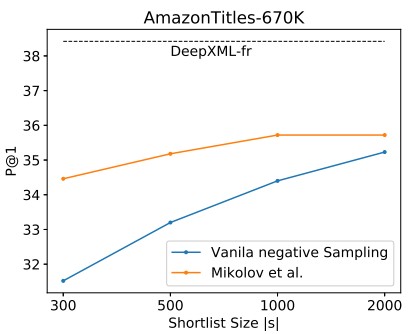 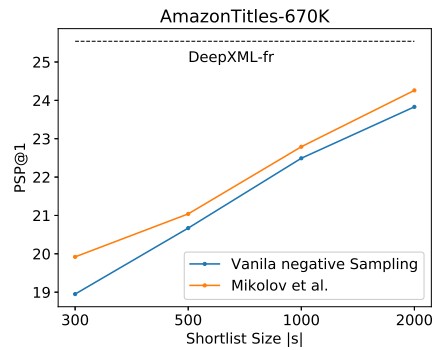

**(a)** Variation of Precision@1 for DeepXML-fr with increase in shortlist size. Black line at the top of the graph indicates Precision@1 for DeepXML-fr without negative sampling

**(b)** Variation of Propensity scored Precision@1 (PS-Precision@1) for DeepXML-fr with increase in shortlist size. Black line at the top of the graph indicates PS-Precision@1 for DeepXML-fr without negative sampling

**Figure 5:** Performance of negative sampling approaches in comparision to 1-vs.-all classifier

### A.3 COMPARISON WITH BoW APPROACHES

DeepXML, DeepXML-RE yields $3 - 4\%$ better accuracy on propensity scored metrics and can be upto $2\%$ more accurate on vanilla metrics. Note that PfastreXML outperform DeepXML and DeepXML-RE on AmazonTitles-3M in propensity scored metrics, however suffers a substantial loss of 10% on vanilla precision and nDCG which is unacceptable for real world applications.

### A.4 EVALUATION METRICS

Performance has been evaluated using propensity scored precision@$k$ and nDCG@$k$, which are unbiased and more suitable metric in the extreme multi-labels setting (Jain et al., 2016; Babbar & Schölkopf, 2019; Prabhu et al., 2018a;b). The propensity model and values available on The Extreme Classification Repository (Bhatia et al., 2016) were used. Performance has also been evaluated using vanilla precision@$k$ and nDCG@$k$ (with $k$ = 1, 3 and 5) for extreme classification.

For a predicted score vector $\hat{\mathbf{y}} \in R^L$ and ground truth vector $\mathbf{y} \in \{0, 1\}^L$:

$$P@k = \frac{1}{k} \sum_{l \in rank_k(\hat{\mathbf{y}})} y_l$$

$$PSP@k = \frac{1}{k} \sum_{l \in rank_k(\hat{\mathbf{y}})} \frac{y_l}{p_l}$$

$$DCG@k = \frac{1}{k} \sum_{l \in rank_k(\hat{\mathbf{y}})} \frac{y_l}{\log(l + 1)}$$

$$PSDCG@k = \frac{1}{k} \sum_{l \in rank_k(\hat{\mathbf{y}})} \frac{y_l}{p_l \log(l + 1)}$$   (2)

$$nDCG@k = \frac{DCG@k}{\sum_{l=1}^{\min(k, ||\mathbf{y}||_0)} \frac{1}{\log(l+1)}}$$

$$PSnDCG@k = \frac{PSDCG@k}{\sum_{l=1}^{k} \frac{1}{\log l + 1}}$$

Here, $p_l$ is propensity score of the label $l$ proposed in (Jain et al., 2016).

### A.5 Literature survey

**Representations:** Deep architectures such as CNN (Liu et al., 2017), MLP (Zhang et al., 2018), and LSTM with Attention (You et al., 2018) have been applied to learn semantically and syntactically rich features. Barring AttentionXML (You et al., 2018), these methods suffer from low accuracy, which is more prominent on tail labels indicating inept document representation for tail labels. However, performance of AttentionXML also degrades for short text documents as discussed in section 4.3.

**Parabel**: Parabel learns a hierarchy over labels to select hardest negatives for each labels and hence bring down the training cost to $O(Nd \log L)$. However, Parabel is designed specifically for sparse BoW features and its performance degrades on low dimensional features.

**Slice**: Negative sampling is a popular approach to reduce training complexity in extreme classification setting. Several strategies have been proposed in literature to select negatives labels for each instance Mikolov et al. (2013); Reddi et al. (2018) or negative examples for each label (Yen et al., 2017; Jain et al., 2019; Prabhu et al., 2018b). Slice approach has been shown to scale to 100 million labels and more accurate than alternate approaches (Yen et al., 2017; Prabhu et al., 2018b) for low dimensional dense features. Slice uses a ANNS to select $O(\frac{N}{L} \log L)$ hardest negative examples and train with selected examples only, thereby reducing the training cost to $O(Nd \log L)$. Slice also brings down the prediction cost to $O(d \log L)$ by evaluating only $O(\log L)$ most probable labels selected from ANNS.

### A.6 Ablation study

Experiments were carried out with several variations of DeepXML where labels were not splitted in head and tail labels. Here are the different configurations:

**DeepXML-f**: This variation refers to a word embedding layer, ReLU non-linearity, Dropout and a fully connected output layer.

**DeepXML-fr**: This variation refers to a word embedding layer, ReLU non-linearity, Dropout, a residual block and a fully connected output layer.

**DeepXML-NS**: This variation refers to a word embedding layer, ReLU non-linearity, Dropout, a residual block and a classifier trained via negative sampling (Mikolov et al., 2013).

**DeepXML-SW**: DeepXML can exploit sub-words features as proposed in (Joulin et al., 2017). This variations refers to DeepXML with sub-word information. Here, character tri-grams were added to the vocabulary in addition to unigrams.

**DeepXML-ANNS**: This variation refers to a word embedding layer, ReLU non-linearity, Dropout, a residual block and a classifier trained via Slice. This version trains only with hardest negatives labels selected via ANNS (Jain et al., 2019). However, the hardest negatives keeps changing as the document representations are being updated. Hence, it requires ANNS graph to be trained multiple time. Here, ANNS graph was updated after an interval of 5 epochs.

**DeepXML-P**: This variation refers to a word embedding layer, ReLU non-linearity, Dropout, a residual block and a shallow tree based classifier proposed in AttentionXML.

**Table 8:** DeepXML and DeepXML-RE can perform on par with Parabel and DiSMEC on full-text documents. Please note that '*' marked algorithms uses slightly different version of the dataset. Values indicated by '-' were not available. Please note that the focus of the paper is on short-text datasets.

| Dataset | Method | P@1 | P@3 | P@5 | N@5 | PSP@1 | PSP@3 | PSP@5 | PSN@5 |
|---|---|---|---|---|---|---|---|---|---|
| Wikipedia-500K | DeepXML | 68.56 | 47.64 | 36.62 | 56.91 | 28.67 | 33.27 | 36.51 | 36.04 |
| | DeepXML-RE | 69.58 | 47.56 | 36.02 | 56.62 | 30.16 | 34.46 | 37.21 | 37.13 |
| | XML-CNN | 59.85 | 39.28 | 29.81 | 46.12 | - | - | - | - |
| | XT | 64.48 | 45.84 | 35.46 | - | - | - | - | - |
| | AttentionXML-1* | 75.07 | 56.49 | 44.41 | 65.77 | 30.05 | 37.31 | 41.74 | - |
| | AttentionXML* | **76.95** | **58.42** | **46.14** | **68.23** | 30.85 | **39.23** | **44.34** | - |
| | SLICE+FastText | 27.98 | 16.61 | 12.11 | 22.69 | 15.04 | 14.61 | 15.17 | 17.59 |
| | DiSMEC | 70.2 | 50.6 | 39.7 | 40.5 | 31.2 | 33.4 | 37 | 37.1 |
| | Parabel | 68.7 | 49.57 | 38.64 | 58.62 | 26.88 | 31.96 | 35.26 | 34.61 |
| | AnnexML | 64.64 | 43.2 | 32.77 | 52.42 | 26.88 | 30.24 | 32.79 | 33.33 |
| | PfastreXML | 59.5 | 40.2 | 30.7 | 28.7 | 29.2 | 27.6 | 27.7 | 28.3 |
| | ProXML | 68.8 | 48.9 | 37.9 | 38 | **33.1** | 35 | 39.4 | **39** |
| Amazon-670K | DeepXML | 44.32 | 39.87 | 36.55 | 40.8 | 30.55 | 33.12 | 35.51 | 32.95 |
| | DeepXML-RE | 44.78 | 40.28 | 36.94 | 41.21 | 30.17 | 33.13 | 35.79 | **33.03** |
| | XML-CNN | 35.39 | 31.93 | 29.32 | 32.64 | 28.67 | 33.27 | 36.51 | - |
| | XT | 42.5 | 37.87 | 34.41 | 38.43 | 24.82 | 28.2 | 31.24 | 28.29 |
| | AttentionXML-1 | 45.66 | 40.67 | 36.94 | 41.35 | 29.3 | 32.36 | 35.12 | - |
| | AttentionXML | **47.58** | **42.61** | **38.92** | **43.5** | 30.29 | **33.85** | **37.13** | - |
| | SLICE+FastText | 33.15 | 29.76 | 26.93 | 30.27 | 20.2 | 22.69 | 24.7 | 22.72 |
| | DiSMEC | 44.7 | 39.7 | 36.1 | 40.5 | 27.8 | 30.6 | 34.2 | 30.7 |
| | Parabel | 44.89 | 39.8 | 36 | 40.36 | 25.43 | 29.43 | 32.85 | 30.71 |
| | AnnexML | 42.39 | 36.89 | 32.98 | 37.04 | 21.56 | 24.78 | 27.66 | 24.76 |
| | PfastreXML | 39.46 | 35.81 | 33.05 | 36.69 | 29.3 | 30.8 | 32.43 | 31.49 |
| | ProXML | 43.5 | 38.7 | 35.3 | 39.7 | **30.8** | 32.8 | 35.1 | 32.6 |

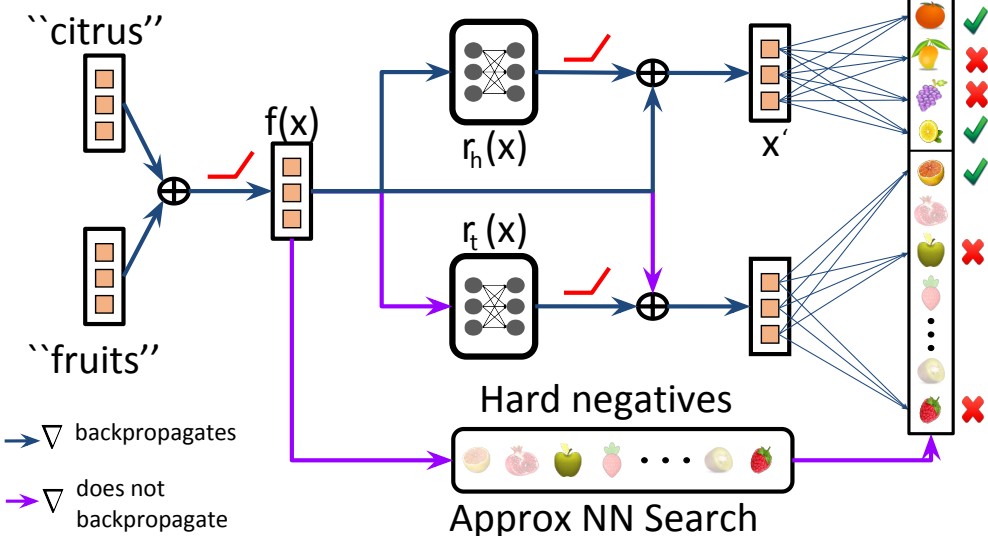

**Figure 6:** An intuitive example with DeepXML architecture. (To be viewed under magnification.)

