# OpenReview forum: "DeepXML: Scalable & Accurate Deep Extreme Classification for Matching User Queries to Advertiser Bid Phrases"
_ICLR.cc/2020/Conference — Reject_

### Official Review · AnonReviewer1 · 2019-10-23
**Official Blind Review #1**

**Rating:** 6

**Review:**

The paper presents a deep learning method for extreme classification and apply it to the application of matching user queries to bid phrases. The main idea is to learn the deep models separately for head and tail labels. Since there is abundant training data for the head labels and transfer the learnt word embeddings for the network for tail-labels, while keeping the head network fixed.

On the positive side, given that there are relatively few successful approaches for deep learning in extreme classification, the main contribution of the paper is towards making an attempt towards this goal.

However, since the paper is mostly empirical in nature and based on algorithmic implementation, the experimental evaluation does not seem quite convincing for the following reasons :

1. Firstly, all the datasets used in the work are private and not publically available. This is quite in contrast to all the various works in this community which use publicly available data and codes.

2. It is not clear why the authors did not to evaluate their approach on the "standard" datasets from the Extreme Classification Repository http://manikvarma.org/downloads/XC/XMLRepository.html. Though it is clear that the focus of the paper is on short text classification, but it is important to evaluate what happens when that is not case. Does the method also works well in longer training/test instance, as there is no immediate reason for it to not work well in that case. Or is it that other methods outperform in that scenario.

3. The performance of the proposed method DeepXML is not significantly better than Parabel. For instance,  on two of the four datasets in Table 1, it gives same predictive performance with order of magnitude less training time and much lower prediction time. This begs the question of utility of proposed approach.

4. Related to above is impact of data pre-processing for different methods. DeepXML seems to use tf-idf weighted word embeddings while other methods use simply BoW representation. It is possible that using simialr data representation or combination with bigrams/trigrams might also improve performance of Parabel and DiSMEC, since it is known from short text classification that using this info can improve performance (https://www.csie.ntu.edu.tw/~cjlin/papers/libshorttext.pdf).

5. Lastly, it is unclear why AttentionXML and DiSMEC are shown to be non-scalable for Amazon3M when they have shown to be evaluated on the bigger version of the same datasets in other works. Also, it might be noted that AttentionXML in the latest version can be combined with shallow trees for efficient scaling.

**Experience Assessment:**

I have published in this field for several years.

**Review Assessment: Checking Correctness Of Derivations And Theory:**

N/A

**Review Assessment: Checking Correctness Of Experiments:**

I carefully checked the experiments.

**Review Assessment: Thoroughness In Paper Reading:**

I read the paper at least twice and used my best judgement in assessing the paper.

---

> ### Author Response · Authors · 2019-11-08
> **Response to Official Blind Review #1 [Part 1/2]**
>
> We thank the reviewer for the constructive criticism and helpful feedback. We would also like to thank the reviewer for recognizing that less than a handful deep learning based approaches have ever been shown to scale to the extreme multi-label classification setting. The reviewer's main criticisms seem to be around the experiments, in which regard we would like to clarify the following:
>
> 1. As mentioned in the abstract and contribution, the dataset and the code will be made publicly available when the paper is accepted. Furthermore, raw text (and labels) for 3 of the datasets namely, AmazonTitles-670K, WikiTitles-500K and AmazonTitles-3M is already publicly available at the Extreme Classification Repository. We have used “titles” from the standard datasets from Extreme classification Repository suited for short text documents.
>
> 2. Thank you for recognizing that focus of the paper is on short text documents where we clearly demonstrate that DeepXML consistently outperforms the current state-of-the-art methods. Our architecture is specifically designed for short text documents, focusing on accuracy and scalability. In particular, for real world applications such as Q2B predictions need to be made in milliseconds on the CPU and therefore expensive architectures as those employed in AttentionXML and XML-CNN are unsuitable.
> Nevertheless, for the sake of completeness and as requested by the reviewer, we have experimented with full text datasets and results are summarized as follows. We have added these results in Table 8 in the supplementary section as well.
>
> A. Amazon-670K
> ————————————————
> Method                 PSP@1        P@1
> ————————————————
> DeepXML-RE        30.17       44.78
> DiSMEC                 27.8         44.7
> Parabel                 25.43       44.89
> AttentionXML       30.29       47.58
> ProXML                 30.8         43.5
> ————————————————
>
> B. Wikipedia-500K
> ————————————————
> Method                 PSP@1        P@1
> ————————————————
> DeepXML-RE        30.16        69.58
> DiSMEC                 31.2          70.2
> Parabel                 26.88        68.7
> AttentionXML       30.85        76.95
> ProXML                 33.1          68.8
> ————————————————
>
> As we can clearly see LSTM/AttentionXML based architecture does provide benefit over DeepXML for long text datasets such as Wikipedia-500K, where a document may contain hundreds (sometimes even thousands) of words.
> However, in our scenario, i.e., short text documents, DeepXML-RE can lead to a gain of 1.0-4.3% in performance (vanilla and propensity scored precision@1) while being 33-42× faster at training than AttentionXML which incorporates an Attention mechanism for each label.
>
> 3. A. In extreme classification, propensity scored metrics are as critical, if not more so, than the vanilla metrics as far as real life applications are concerned. DeepXML/DeepXML-RE is able to consistently outperform the current state-of-the-art approaches such as DiSMEC, Parabel, and AttentionXML on vanilla metrics and even more so on propensity-score metrics. The results (Table 1 & Table 2) can be summarized as: i) Vanilla precision/nDCG: DeepXML-re can be 2.91%, 2.76%, and 1.08% more accurate than Parabel in terms of precision@1 on WikiSeeAlsoTitles-500K, WikiTitles-500K, and AmazonTitles-670K respectively, while being comparable (~0.75% worse) to Parabel on AmazonTitles-3M. ii) Propensity scored precision/nDCG: DeepXML-RE can be 2.39%, 2.21%, 4.21%, and 4.11% more accurate than Parabel in terms of PSP@1 on WikiSeeAlsoTitles-500K, WikiTitles-500K, AmazonTitles-670K, and AmazonTitles-3M respectively. iii) Additionally, results on the Q2B-3M dataset further strengthen the argument for DeepXML, where DeepXML-re has been found to be 19% more accurate in terms of P@1 and 21.71% more accurate on PSP@1, relative to Parabel.
> B. Regarding the utility of DeepXML, we would like to clarify that Q2B-3M is a highly impactful real world application being used by millions of users around the world (though, unfortunately, personal user queries cannot be released publically due to privacy and other concerns). When deployed in production in a live flight, DeepXML was found to benefit many users and advertisers even though our search engine already has a large ensemble of state-of-the-art techniques for this task. In particular, DeepXML added more than 67 million new good quality predictions over the ensemble. Furthermore, for 8.6% of queries, DeepXML was able to match the query to a good quality bidded keyword and show an ad whereas none of the algorithms in the production ensemble could. Simultaneously, DeepXML increased the quality of predictions by 2.9% over the ensemble. As a result, this increased revenue per thousand queries by 1.64% which is highly significant given the overall volume. Thus, to reiterate, DeepXML can lead to large gains in performance for real world applications impacting millions of users.
>
> [Continued below]

---

> > ### Author Response · Authors · 2019-11-08
> > **Response to Official Blind Review #1 [Part 2/2]**
> >
> > 4. a) In the extreme classification community TF-IDF based BoW features are widely used. We would like to clarify that we have also used the same TF-IDF based BoW features for training/evaluating both DiSMEC/Parabel and DeepXML/DeepXML-RE. Please note that the dataset statistics are already included in Table 4 in the appendix.
> > b) Bi-gram and tri-gram features may lead to improvements in BoW based methods for certain applications. However, bi-gram and tri-gram features can also lead to an increase in training time, prediction time and model size due to increased number of features. We had experimented with bi-grams and tri-grams and they did not lead to any significant gain in our experiments. Please refer to the following table which demonstrates results with different numbers of bi-grams for Parabel on AmazonTitles-670K. Here, both uni-grams and bi-grams were included in the vocabulary and hence the features. It should be mentioned that here TF-IDF features are used, similar to the rest of the paper.
> >
> > ——————————————————
> > #unigrams    #bigrams    P@1    PSP@1
> > ——————————————————
> > 66666             0                  38.00    23.10
> > 66666             10000          37.76    22.86
> > 66666             50000          37.95    22.93
> > 66666             100000        38.16    23.12
> > ——————————————————
> >
> > 5. a) DiSMEC could potentially be scaled to AmazonTitles-3M, however it would require roughly 2 weeks to train on a machine with the following configuration: Intel(R) Xeon(R) CPU E5-2673 v4 @ 2.30GHz (64 cores).
> > b) AttentionXML could also be potentially scaled to AmazonTitles-3M, however, it would require 150+ hours on a single P40 GPU card.
> > These training times are an order of magnitude higher than DeepXML and are prohibitive for real world applications. Nevertheless, as requested by the reviewer, we are running these experiments and will update the paper with the numbers as soon as they are available.
> > c) We would also like to clarify that we have used the most recent version of AttentionXML (i.e., with shallow trees). The code for this was provided by the authors.

---

### Official Review · AnonReviewer2 · 2019-10-26
**Official Blind Review #2**

**Rating:** 3

**Review:**

This paper considers extremely multi-label learning (XML) where the label size is very large . It aims to improve the accuracy as well as the scalability of XML algorithms, especially for short text inputs. The accuracy for XML with short text inputs can be significantly improved using deep learning representation than using TFIDF features. This paper proposes several tricks to handle the issue for efficiently learning both neural network parameters and classification weights of extremely large number of labels. The proposed method borrowed ideas from several previous literature, and is mainly based on SLICE, where a pre-trained fixed deep learning representation for the inputs are used with ANNS graph for labels to improve the scalability. The main difference is that instead of using a fixed input embedding, the proposed method learns the word embedding parameters via a set of head labels. The remaining labels are then trained using SLICE with fixed word embeddings from the learned word embedding model.

Overall the paper tackles the problem well. And the empirical results show improved results. However, I don't think this paper is ready for publication due to the following concerns.

1. My main concern is that the proposed method seems to be a combination of a number of tricks. This makes the overall algorithm/model very complicated and introduces a lot of hyper-parameters, for example, head label portion, L-h', c, beta, s neural network hyper-parameters and so on. Hence, it will be hard to be used in real applications.

2. Another concern is about the experiments.
    a. The most significant improvement of the proposed method over existing method happens in the private dataset, Q2B-3M, which can't be reproduced.
     b. On the public datasets, DeepXML seems to show good results on small datasets, WikiSeeAlsoTItles-350K and WkipediaTitle-500K, while on large datasets, DeepXML performance is close to the existing methods.
     c. The largest label size in the experiments is 3M. SLICE can be scaled up to 100M labels.

3. The writing and the organization of this paper needs to be improved.
    a. Some notations are not clearly defined. For example, L_h in Line 6 and X' in Line 9 on Page 5.
    b. Several method names are not defined. For example, DeepXML-fr, AttentionXML-l, Sliced-CDSSM, DeepXML-SW, DeepXML-f. I have to guess what they are.
    c. The last two paragraphs on Page 4 seems to be related work, while there is a section called "Related work".

Other minor comments:
1. It seems it is not stated how Beta is set.
2. I am wondering if it's true that the shorter the input text is, the better improvement over non-deep-learning methods DeepXML can achieve.
3. In the first paragraph of Sec 3.1, it is mentioned "clustering just the top 3 labels into 300 clustering". Why choose 3 and 300? Are these numbers used for all datasets?

**Experience Assessment:**

I have published one or two papers in this area.

**Review Assessment: Checking Correctness Of Derivations And Theory:**

N/A

**Review Assessment: Checking Correctness Of Experiments:**

I assessed the sensibility of the experiments.

**Review Assessment: Thoroughness In Paper Reading:**

I read the paper at least twice and used my best judgement in assessing the paper.

---

> ### Author Response · Authors · 2019-11-08
> **Response to Official Blind Review #2 [Part 1/2]**
>
> We thank the reviewer for the constructive criticism and helpful feedback. We would like to take this opportunity to address the expressed concerns:
>
> 1. A. we would like to clarify that Q2B-3M is a highly impactful real world application being used by millions of users around the world (though, unfortunately, personal user queries cannot be released publically due to privacy and other concerns). When deployed in production in a live flight, DeepXML was found to benefit many users and advertisers even though our search engine already has a large ensemble of state-of-the-art techniques for this task. In particular, DeepXML added more than 67 million new good quality predictions over the ensemble. Furthermore, for 8.6% of queries, DeepXML was able to match the query to a good quality bidded keyword and show an ad whereas none of the algorithms in the production ensemble could. Simultaneously, DeepXML increased the quality of predictions by 2.9% over the ensemble. As a result, this increased revenue per thousand queries by 1.64% which is highly significant given the overall volume. Thus, to reiterate, DeepXML can lead to large gains in performance for real world applications impacting millions of users.
> Furthermore, [for Q2B-3M] DeepXML used the default hyper-parameters determined from small and moderate-sized datasets, mitigating the need for any hyper-parameter tuning.
>
> B. As included in Table 5 (in the appendix), DeepXML takes 3 tunable hyper-parameters, namely the threshold to split labels, beta, and the learning rate for DeepXML-h. Other parameters such as the shortlist size (|s|) and the learning rate for DeepXML-t etc. have been chosen to have the same values for all datasets. Please note that only the learning rate for DeepXML-h (only 1 out of 3 tunable parameters) requires re-training the head network, i.e. DeepXML-h, in order to search for the optimal value.
> i. The proposed approach splits labels based on frequency [Section 4.2 and Section A.2 (in the appendix)]. The frequency threshold for splitting the labels is chosen while keeping the following points in mind: i) The number of head labels must not grow beyond 200K labels. ii) Most of the words/tokens should be covered in the vocabulary for head labels. Please note that the threshold is chosen strictly based on the aforementioned criteria and hence we train DeepXML only for the chosen threshold frequency, i.e., no re-training required.
> ii.  The beta parameter which controls the weightage of the classifier score and the ANNS score doesn’t have any impact on the training time. Beta is chosen post-prediction in order to achieve the best precision on the validation set. The impact of beta is already covered in Fig. 3 [in the appendix].
>
> C. It seems that we have inadvertently conveyed the misconception that our paper just learns the features on the head and then runs Slice on these fixed features. Doing this in our experiments lead to an accuracy drop of 1-2% and 3x decrease in efficiency. Thus the focus of our paper is to learn apt feature representation and to address these limitations of Slice so as to actually increase the accuracy and improve the efficiency. In particular, the following limitations of Slice were addressed: a) Slice would have required to re-trains ANNS graph multiple times as the features are learnt for DeepXML-t as well. DeepXML addresses this limitation by using pre-residual features to train ANNS and post-residual features to train the classifier. b) Slice would have required to sample ~3x labels leading to 3x increase in cost of classifier during training and prediction.
>
> DeepXML makes principled design choices which were required to achieve state-of-the-art accuracy and scalability. The design choices of splitting labels, feature representations, and classifier are motivated in the “DeepXML, FastText & Slice” subsection [Page 4] and backed by empirical results in Tables 1 and 2.
>
> [Continued below]

---

> > ### Author Response · Authors · 2019-11-08
> > **Response to Official Blind Review #2 [Part 2/2]**
> >
> >
> > 2. A. Please refer to 1 (A).
> >
> > B. In extreme classification, propensity-scored metrics are as critical, if not more so, than the vanilla metrics as far as real life applications are concerned [Accuracy subsection in Section 2 (Related work)]. DeepXML/DeepXML-RE is able to consistently outperform the current state-of-the-art approaches such as DiSMEC, Parabel, and AttentionXML on vanilla metrics, and even more so on propensity-scored metrics. Specifically, DeepXML-RE can be 2.39%, 2.21%, 4.21%, and 4.11% more accurate than Parabel in terms of PSP@1 on WikiSeeAlsoTitles-500K, WikiTitles-500K, AmazonTitles-670K, and AmazonTitles-3M respectively [Table 1; Page 7].
> >
> > C. We agree with the reviewer that Slice can scale to 100 million labels, however with pre-trained features only. As demonstrated in Table 3 [Page 8], pre-trained features with Slice classifier are unable to perform on-par with DeepXML. Unfortunately, Slice loses both accuracy and scalability if features are to be learnt alongside the classifier. Firstly, Slice needs to train the ANNS graph multiple times when features are being constantly updated. Secondly, when the features are pre-trained, the loss function decomposes over the labels and therefore Slice is able to train each of the classifier in parallel [Eq. 1 & 2 in Jain et al. 2019]. This property is no longer true when features are learnt and therefore the source of parallelism and hence efficiency goes away.
> > Specifically, Slice with feature training [referred to as DeepXML-ANNS in Table 3] was found to be ~2.3% less accurate relative to DeepXML-RE.
> >
> > 3. A & B. We would like to thank the reviewer for pointing out the unclear aspects. We have updated the paper to clearly define notation such as X’, L_h, and method names [Section 5 and Section A.6 (in the appendix)].
> > C. [Last 2 Paragraphs on Page 4] discuss the highly relevant papers such as fasttext, Slice, etc. in detail whereas the Related Work section covers the relevant approaches at a higher level. Additionally, the same paragraph presents our motivation behind the design choices.
> >
> > 4. a) Please refer to setting beta in hyper-parameters [point 1 (B) in response].
> > b) DeepXML does excel over non-deep-learning based methods on especially on short text. However, the accuracy also depends significantly on the following factors: i) number of labels, ii) feature distribution, iii) training points, and iv) label distribution.
> > c) After carefully analyzing the label distribution of the WikiTitles-500K dataset we observed that 3 labels such as “living people” occurred in tens of thousands of documents and it was sufficient to use multiple representatives for these three labels only. Additionally, the number of clusters were chosen empirically in order to improve recall values [Line 8; Para 2; DeepXML-h; Section 3.1].
> > This problem is specific to the datasets where some labels are highly diverse [Para 2, DeepXML-h, Page 5] in nature and that is the case with WikiTitles-500K only [for e.g. “Living people” tag]. Ideally, one could cluster data points and use multiple representatives for every label in L_h. However, it leads to an increase in training time without significant gain in accuracy.

---

### Official Review · AnonReviewer3 · 2019-10-29
**Official Blind Review #3**

**Rating:** 6

**Review:**

This paper introduces a new algorithm for extreme multi-label classification. The focus of this paper is on being able to handle short documents with all experiments focussed on matching user queries to advertiser bid phrases. The key novelty in this paper is to split the labels into two buckets: head labels and tail labels. The model learns word embeddings on the head labels + a classifier on top of those embeddings. For the tail labels, the embeddings from the head labels are used as the input for another classifier which is trained on only the tail.

My thoughts on the paper:
- I really like the paper writeup; its succinctness (in most sections, some comments below).
- Small nit: why choose head labels as 0.1L? I would have expected a more natural choice to be based on frequency?
- Figure 1 (and explanation in section 3.1): I understand the head setup completely (although it seems to be missing the ANNS). For the DeepXML-t part, I am not very clear from the picture nor the explanation how the ANSS feeds into the weights and how that leads to the label \hat{y}+{clf-t}?
- Section 3.1, DeepXML-h section: the bit after "Additionally ..." until "DeepXML-t" section is unclear. I think this needs to be explained better.
- I might have missed it but is PSP metric defined explicitly somewhere?
- The experiment section contains lots of baseline comparisons; unfortunately not all on publicly available datasets.
- The paper uses a large number of aconyms which are defined sometimes after their first use, sometimes never: i.e. PLT, ANNS.

**Experience Assessment:**

I do not know much about this area.

**Review Assessment: Checking Correctness Of Derivations And Theory:**

I assessed the sensibility of the derivations and theory.

**Review Assessment: Checking Correctness Of Experiments:**

I assessed the sensibility of the experiments.

**Review Assessment: Thoroughness In Paper Reading:**

I read the paper at least twice and used my best judgement in assessing the paper.

---

> ### Author Response · Authors · 2019-11-08
> **Response to Official Blind Review #3**
>
> We would like to thank the reviewer for the constructive feedback and kind words regarding the writing. We would like to take this opportunity to address the expressed concerns:
>
> 1. 0.1L is used to assert that less than 10% of the labels are treated as head labels by DeepXML for the large datasets such as AmazonTitles-3M and Q2B-3M. The proposed approach indeed splits labels based on frequency [Section 4.2 and Section A.2 (in the appendix)]. The frequency threshold for splitting the labels is chosen while keeping the following points in mind: i)  The number of head labels must not grow beyond 200K labels. This is important for practical purposes as the head network, i.e. DeepXML-h, is trained with a fully connected layer (classifier) and increasing the labels can lead to increase in training time. For instance, DeepXML-fr, which consider all labels as head labels, can be 3-5x slower to train than DeepXML [Table 1; Page 7]. ii) Most of the words should be covered in the vocabulary for head labels. Note that word embeddings are not updated for the tail network. Hence, the network needs to learn word embeddings for all words in the vocabulary from only the head labels. Additionally, the exact number of head and tail labels for various datasets are included in Table 5 in the appendix.
>
> 2. Inference in the tail network, i.e. DeepXML-t, relies on following components: i) The ANNS graph which returns the indices and scores (cosine similarity) corresponding to the |s| most probable labels for a novel test point, ii) A classifier which is evaluated (dot product) only on the most probable labels returned by the ANNS graph. Hence, the final score is the combination of the ANNS score, i.e. $\hat{y}_{anns-t}$, and the classifier score $\hat{y}_{clf-t}$, as included in Equation 1 in Section 3.2. This approach brings down the classifier cost to $O(d log(|L_t|)$ + d |s|) from $O(d |L_t|)$. Please note that $|s|$ is kept as 300 for all the datasets.
>
> 3. The head network, i.e. DeepXML-h, is trained with a fully connected output layer. However, in order to meet the low latency constraint of real life applications an ANNS structure is trained post-training (for DeepXML-h) which brings down the inference cost of the classifier to $O(d \log(|L_h|) + d |s|)$ from $O(Nd |L_h|)$. [Line 7; Para 1; DeepXML-h; Section 3.1].
> However, trivially training an ANNS leads to poor recall values on head labels, i.e., some of the true labels didn't appear in the ANNS shortlist [Para 2 in 3.1 (DeepXML-h)]. Please note that if a label (say $l_i$) doesn't appear in the shortlist (for a novel test instance $x_j$), then the classifier will not be evaluated for $l_i$ and hence the network will not be able to predict it for $x_j$. This problem mainly occurs for labels with highly diverse contexts such as "Living People" in WikiTitles-500K. DeepXML tackles this problem by allowing multiple representatives for the aforementioned labels resulting in a 5% increase in recall@300 and 6% in precision@300 with a shortlist of size 300 [Line 8; Para 2 in 3.1 (DeepXML-h)].
>
> 4. PSP@k [Jain et al. 2016] is a standard metric used in extreme classification. We have added the definitions in A.6 in the supplementary section for completeness. Please note that predicting rare tail labels accurately is much more rewarding than predicting common and obvious head labels in extreme multi-label learning [Please refer to the "Tail labels" subsection under the Introduction (Section 1)]. Hence, PSP@k, which focuses more on tail labels is critical to real word applications such as matching user queries to advertiser Bid Phrases, i.e., Q2B-3M.
>
> 5. As mentioned in the abstract and contribution, the dataset and the code will be made publicly available when the paper is accepted. Furthermore, raw text (and labels) for 3 of the datasets namely, AmazonTitles-670K, WikiTitles-500K, and AmazonTitles-3M is already publicly available at the Extreme Classification Repository. We have used "titles" from the standard datasets from Extreme classification Repository suited for short text documents.
>
> 6. Some of the acronyms such as PLT (Jasinska et al., 2016) and ANNS (Jain et al., 2019) are taken from the respective papers. We have defined all acronyms on first use. Thanks for pointing this out.

---

### Decision · Program_Chairs · 2019-12-19

**Decision:**

Reject

**Comment:**

The paper proposes a new method for extreme multi-label classification. However, this paper only combine some  well known tricks, the technical contributions are too limited. And there are many problems in the experiments, such as the reproducibility, the scal of data set and the results on well-known extreme data sets and so on. The authors are encouraged to consider the reviewer's comments to revise the paper.